# Toward a Value-Based Therapy Recommendation Model

**DOI:** 10.3390/healthcare11162362

**Published:** 2023-08-21

**Authors:** Zhang Liu, Liang Xiao

**Affiliations:** School of Computer Science, Hubei University of Technology, Wuhan 430068, China; liuzhang@hbut.edu.cn

**Keywords:** patient values, clinical decision making, personalized therapy, recommendation model, argumentation

## Abstract

Patient value is an important factor in clinical decision making, but conventionally, it is not incorporated in the decision processes. Clinical decision making has some clinical guidelines as a reference. There are very few value-based clinical guidelines, but knowledge about how values affect decision making is mentioned in some scattered studies in the literature. We use a literature review method to extract evidence and integrate it as part of the decision-making model. In this paper, a value-based therapy recommendation comprehensive model is proposed. A literature analysis is conducted to collect value-based evidence. The patients’ values are defined and classified with fine granularity. Categorized values and candidate therapies are used in combination as filtering keywords to build this literature database. The literature analysis method generates a literature database used as a source of arguments for influencing decision making based on values. Then, a formalism model is put forward to integrate the value-based evidence with clinical evidence, and the literature databases and clinical guidelines are collected and analyzed to populate the evidence repository. During the decision-making processes, the evidence repository is utilized to match patients’ clinical information and values. Decision-makers can dynamically adjust the relative importance of the two pieces of evidence to obtain a treatment plan that is more suitable for the patient. A prototype system was implemented using a case study for breast cancer and validated for feasibility and effectiveness through controlled experiments.

## 1. Introduction

### 1.1. Conventional Methods for Clinical Decision Making

Clinical decision making is a complex process that involves numerous factors, including clinical evidence, patient values, and healthcare professionals’ expertise [1]. Clinical decision support systems (CDSSs) have been developed to aid healthcare providers in making informed decisions. However, traditional CDSSs have limitations in considering patient values, which are crucial in clinical decision making. These limitations can lead to a mismatch between the treatment plan and the patients’ values and preferences, leading to compromised health outcomes.

The recommendation of treatment plans has traditionally been explored in the realm of representing and interpreting clinical guidelines. John Fox and his colleagues proposed PROforma, which is a formalism that offers an evidence-based and objective tool for selecting treatment plans and providing decision support. Its main goal is to empower clinicians with optimal treatment recommendations [2]. Systems based on this formalism have the capability to furnish doctors with tailored treatment recommendations, taking into account patients’ conditions, medical history, and examination results through computer-interpretable representation.

In addition to PROforma, several studies have explored alternative methods and techniques for recommending treatment plans. Romina et al. examined the interpretation and utilization of clinical practice guidelines or recommendations [3]. Some research focuses on recommending treatment plans through the utilization of clinical guidelines or expert knowledge. For instance, Parikh et al. developed guidelines for diagnosing, clinically assessing, treating, and managing patients with mitochondrial disease [4]. Domain et al. proposed a decision support system for breast cancer treatment based on data-mining technologies and clinical practice guidelines. They discussed the system’s implementation, application, and evaluation [5]. In collaboration with John Fox, we have proposed a systematic approach for representing argumentation, recommendation, and explanation in clinical decision support [6].

### 1.2. Studies on Patient Values and Their Relationships with Decision Making

Patient values refer to the individual beliefs, preferences, desires, and priority considerations that patients have regarding their habits, health, and medical decisions. Healthcare professionals need to take into account not only clinical evidence and best practices but also the patient’s values when making medical decisions. Patient-centered medical decision making is crucial because certain treatment options that threaten the patient’s valued beliefs may directly reduce the patient’s adherence to the treatment plan, leading to suboptimal treatment outcomes. Berry et al. outlined six categories of patient values, including activities, abilities, possessions, principles, emotions, and relationships [7]. These values reflect patients’ perspectives and evaluations of their preferred activities, functional abilities, material possessions, guiding principles, emotional well-being, and social relationships. These factors significantly impact medical decision making and the selection of treatment plans.

There is a growing trend of incorporating patient values into Clinical Decision Support Systems (CDSSs) using methods such as Patient-Reported Outcome Measures (PROMs) and Shared Decision Making (SDM) [8,9]. PROMs offer insights into patients’ values and preferences through standardized questionnaires. Torenholt et al. introduced recontextualisation work revealing nurses’ efforts in recontextualizing PRO data [10]. SDM enables collaborative decisions considering treatment options, risks, benefits, and patients’ values. Epstein and Street extensively discussed the significance of patient-centered care [11]. They argue that it better addresses patients’ needs and values, enabling them to participate more effectively in medical decision making. Curtis et al. proposed guidelines to assist healthcare providers in gaining a better understanding of patients’ values and needs [12].

Numerous studies have investigated the integration of values into medical decision support systems. Wherton et al. developed a value-based medical decision support system that aids doctors in comprehending patients’ values and needs [13]. Liu et al. proposed a method to integrate clinical knowledge and patient preferences into an integrated knowledge graph [14]. In our previous work, we presented a model that utilizes clinical experience data as evidence to support patient-oriented decision making [15].

NICE clinical guidelines mention treatment side effects that may challenge patient values but lack standardized analysis [1]. Integrating patient values with CDSSs leads to more personalized treatment plans enhancing patient satisfaction [8,9]. However, the scattered nature of patient-reported PROMs hinders systematic value-based evidence for personalized treatment recommendations.

In summary, while certain studies have introduced the concept of patient values into clinical decision making and developed models that incorporate patient values, there remains a lack of systematic utilization of values to establish concrete evidence of their influence on medical decision making.

### 1.3. Literature Analysis Methods

Literature analysis is a widely adopted research method that facilitates the systematic collection, organization, evaluation, and synthesis of the literature, enabling the exploration and examination of the current state, trends, and issues within a specific research field. This methodology finds applications in various domains, including medicine, social sciences, and education. For instance, Porr et al. employed a literature analysis to investigate the ethical challenges confronted by nurses in long-term care facilities when providing care for individuals with dementia [16] Hsieh et al. utilized a literature analysis to compare and contrast three qualitative content analysis methods, thoroughly discussing their respective advantages, disadvantages, and appropriate usage scenarios [17]. In another study, Taremwa et al. conducted a comprehensive literature analysis to quantitatively analyze the number of publications, themes, and authors concerning malaria vector control and drug resistance in malaria vectors [18].

There may exist implicit mapping relationships between treatment plans and values, which are often not explicitly addressed in clinical guidelines. It is hypothesized that these relationships can be supplemented by analyzing the clinical decision literature using a literature analysis approach, incorporating relevant references to inform the decision-making process. Patient values can be influenced by diverse factors, including cultural background, beliefs, social environment, educational background, and personal experiences. By systematically collecting, organizing, and analyzing the relevant literature, a deeper understanding of patient needs and values can be attained, facilitating the development of treatment plans that better align with patient requirements. Through a literature analysis, the objective is to establish a foundation of arguments centered on patient values. Table 1 presents the key terms identified for different categories of values, while Table 2 presents the key terms for candidate treatments. These two sets of keywords are integrated and utilized as filters to retrieve the relevant literature from the database, thereby enhancing the retrieval of the literature pertinent to the identified keywords.

#### Research Goals

We are committed to integrating objective clinical knowledge with patient values into the clinical decision-making process to find the most suitable treatment plan for patients. We use a literature analysis method to extract evidence of the impact of values on decision making from the literature and, using scientific research methods, integrate this value evidence with relevant clinical data into a comprehensive evidence base to guide patient clinical decision making. The comprehensive evidence base provides interpretable treatment recommendations and has traceability. We recognize that due to the diversity of patient values and diseases, the resulting evidence base may have certain limitations. We follow the following standards and ideas: In the research process, we focus on a literature analysis related to breast cancer, looking for key sentences that influence decision making by values and forming an evidence base for decision making influenced by values. By analyzing the structure of this evidence base and the clinical decision-making evidence base, and using matching algorithms to match patient information with evidence, we will provide treatment options that integrate patient values.

We categorize the objectives of this study into the following three goals:

Goal 1: Summarize a knowledge base by searching for research on the impact of values on decision making in the current clinical literature (Section 2.3).

Goal 2: Construct a unified model that can accommodate both clinical evidence and value-based evidence to provide systematic support for decision-makers (Section 2.4, Section 2.5 and Section 2.6).

Goal 3: Evaluate the feasibility and effectiveness of a unified model for decision-making evidence (Section 3.1, Section 3.2 and Section 3.3).

## 2. Materials and Methods

### 2.1. An Overview of the Model

Due to the current clinical guidelines’ limited consideration of patient values and the lack of connection between patient values and systematic clinical decision making, our goal is to integrate the clinical evidence model and the value-based evidence model into a comprehensive evidence framework. We have taken advantage of the research features of both review articles and scientific papers. This involves extracting evidence of patient values from the literature and objective clinical evidence from clinical guidelines, merging them into a unified evidence repository. This evidence repository will encompass both clinical knowledge and value-based knowledge. The current clinical guidelines and literature summaries are shown in the Box 1 below.
Box 1Excerpts from NICE clinical guidelines.The NICE guideline suggests that when a patient’s indicator is ER+, both chemotherapy and endocrine therapy can be offered as treatment options. According to the medical literature, when a patient’s indicator is ER+ and the patient places more emphasis on treatment duration, chemotherapy may be more advantageous than endocrine therapy.

The model overview, which encompasses the influence of values on decision making, is depicted in Figure 1, and the process unfolds as follows:Collecting clinical evidence and conducting model analysis (detailed in Section 2.2): In the initial phase, we gathered objective clinical knowledge from NICE clinical guidelines, encompassing disease names, associated symptoms, diagnoses, and treatment plans. The interrelationships among these entities are outlined in the NICE clinical guidelines. Based on the analysis of these entities and relationships, we formulated a model for clinical evidence. Subsequently, the clinical knowledge acquired from NICE underwent formatting and decomposition using the proposed approach. To facilitate computer recognition and the processing of clinical evidence, we devised a hierarchical structure to represent the clinical evidence in the RDF format.Collecting value evidence (literature) and developing a model (detailed in Section 2.3): We obtained research papers pertaining to diseases and values from PubMed. To illustrate the process, we employed a combination of keywords, specifically ’breast cancer’, along with the 29 values listed in Table 1, to retrieve 2506 relevant papers. After eliminating duplicates, we were left with 2487 unique papers. Subsequently, we augmented the screening process by incorporating the treatment plans provided by NICE, as presented in Table 2, as additional screening keywords. This refinement resulted in a final set of 341 papers encompassing diseases, treatment plans, and values. These 341 papers were then categorized into 27 groups based on the combination of treatment plans and organized into a literature database. Utilizing a manual reading method, we meticulously reviewed the papers and extracted key content. To effectively capture the manifestation of value influence on decision making within the literature and establish a hierarchical relationship among entities, we devised a model for value evidence.Constructing a unified model for decision evidence (UMDE) (detailed in Section 2.4): To ensure that the value evidence extracted from the literature is accessible for patient decision making, it is crucial to integrate the clinical evidence model and the value evidence model into a unified framework. Hence, we propose the UMDEw w. UMDE incorporates the structural characteristics of both the clinical evidence and value evidence models and leverages the extracted value influence from the literature as an incremental factor in clinical decision making, thereby facilitating subsequent patient decision-making processes.Establishing population-based value pre-configuration (detailed in Section 2.5): During our investigation of patient values, we have observed a strong correlation between the characteristics of the patient population and their prioritized values. For instance, occupation plays a significant role in individuals’ lives, and the values associated with different occupations often align with those valued by patients. Moreover, most patients prefer treatment plans that do not disrupt their occupational routines. To capture this relationship, we have collected extensive occupational information through occupational classification. Leveraging statistical knowledge, we have linked these occupations to medically relevant values and assigned them predefined weights. This approach aims to facilitate the model in providing essential values while exploring patient values in subsequent analyses.Performing runtime clinical data and value elicitation, and VMDE application with weight adjustment (detailed in Section 2.6 and Section 3.1): Clinical symptoms and value information are acquired from patients through the utilization of a problem-guided interview method. The collected clinical information encompasses disease type, symptom severity, disease progression, treatment history, and other relevant details. Moreover, understanding the patients’ personal values is crucial, including their values pertaining to activities, possessions, principles, emotions, relationships, abilities, and the 29 values listed in Table 1. The predefined values outlined are employed to map and analyze the patients’ objective conditions and value inputs. The resulting degree of patient value preference and objective situation, obtained by the current model, are then outputted and provided to the patients for data refinement and verification. The patients’ fine-tuning behavior regarding the data is recorded within the model to enable the automatic adjustment of predefined values, rendering them more realistic. Clinical symptom information obtained from patients is compared with symptom details in treatment plans and ranked based on severity and impact, utilizing predefined weights. These weights are subsequently used for plan recommendation and weight calculation in subsequent stages. The ranking of objective symptom information, based on weight, is integrated with evidence-based decision making rooted in values. Relevant information that aligns with the evidence is extracted, and objective weights are consolidated to determine the final weight for each recommended plan. Different treatment plans are then ranked and recommended based on the degree of influence and weight assigned to distinct values, thereby assisting doctors in formulating personalized treatment plans that align with patient requirements. For detailed algorithms, please refer to Section 2.6.

Through these sequential steps and processes, patients’ personalized needs and value factors can be integrated into clinical decision making and treatment plan selection, adhering to a value-based decision-making approach. This, in turn, assists doctors in comprehensively understanding patients’ needs and expectations as well as developing highly personalized treatment plans to meet their individual requirements.

### 2.2. Collecting and Modeling Clinical Evidence

In this section, a method has been employed to extract objective knowledge evidence from clinical guidelines [19]. The specific steps involved in this approach will now be provided in detail. By systematically extracting objective knowledge evidence from clinical guidelines, this section aims to contribute to the development of a comprehensive and reliable knowledge base.

Based on previous work, the treatment option analysis for breast cancer was specifically chosen from the NICE guidelines. In order to extract objective knowledge, a combination of manual reading and information extraction tools was utilized, as illustrated in Figure 2. The extracted objective knowledge encompasses crucial information about the disease, treatment options, and associated symptoms. To ensure a structured representation of this knowledge, a hierarchical resource description framework (RDF) format was meticulously designed, as depicted in Figure 3. The RDF consists of two distinct parts: the first part focuses on the treatment options, while the second part elaborates on the weights assigned to objective symptoms for each treatment option. This structured representation enables a comprehensive and organized understanding of the extracted knowledge, facilitating subsequent analysis and decision-making processes.

Patient information is effectively matched with the symptoms stored in the RDF, thereby activating the corresponding arguments within the database. Each argument is equipped with a “support-type” attribute, which signifies its stance in relation to the treatment option. When the attribute value is “for”, it indicates that the activated argument supports the corresponding treatment option, and the specific weight is specified within the “weight” attribute. Conversely, when the attribute value is “against”, it signifies that the activated argument opposes the treatment option, and the weight is derived from the negation of the value provided in the “weight” attribute. Through the process of matching and activating the objective symptoms using the RDF file, the weights and argument information for each treatment option are effectively obtained, enabling a comprehensive evaluation and comparison of different treatment options based on their corresponding evidence.

This systematic approach serves as a valuable resource, empowering medical professionals and decision-makers to make well-informed choices when considering various breast cancer treatment options. The structured RDF representation of the extracted knowledge further enhances the clarity and comprehensibility of the information. For a more detailed visualization and understanding of the RDF structure, please refer to Figure 3.

### 2.3. Value-Based Medical Decision Making and Treatment Plan Selection

We adopted a literature analysis method to investigate the impact of diverse values on medical decision making and treatment plan selection. It is crucial to emphasize that the automatic keyword filtering from the repository is conducted using code, whereas the extraction of value evidence relies on a manual reading of the literature content. Before commencing the literature screening process, it is essential to establish a dedicated keyword database to identify relevant literature pertinent to our research objectives.

We have expanded on prior research [7] by enhancing the six categories of values, specifically activities, abilities, relationships, emotions, principles, and possessions, through the introduction of more nuanced sub-values. These refined values can fall into multiple categories, and their interrelationships are outlined in Table 1.

The classification and categorization of values aim to enhance our understanding of individuals’ experiences and needs in medical treatment as well as the focus and priorities of medical institutions and practitioners when it comes to patient care. For the fine-grained values, we need to classify and sort them to facilitate literature screening and analysis. It should be noted that certain fine-grained values may fall into multiple categories. Thus, we need to organize and classify the relationships between these values to enhance the literature screening and analysis.

As an illustrative example, our study specifically targeted breast cancer, and we extracted 14 treatment plans mentioned in the NICE guidelines to serve as keywords for subsequent literature screening. These keywords function as criteria for matching and identifying literature pertaining to breast cancer treatment plans. The specific keywords utilized in this context are provided in Table 2. By employing these keywords as screening criteria, we can identify the literature that specifically addresses breast cancer treatment plans, allowing for further analysis of the values and concerns associated with them.

It was observed that utilizing a combination of disease and a single value as search keywords had the potential to retrieve articles multiple times, especially if they contained multiple values. To mitigate this issue, a filtering process was implemented to eliminate articles with completely duplicated titles and abstracts. This meticulous filtering process resulted in a final selection of 2487 unique articles, ensuring the inclusion of diverse and distinct literature for analysis.

The primary objective of our research is to establish meaningful connections between values and relevant treatment plans. To achieve this, the initial pool of 2487 articles underwent a meticulous screening process, specifically targeting those articles that mentioned the specific treatment plan keywords listed in Table 2. Through this rigorous screening, a subset of 341 articles was identified from the original 2506 articles, where these selected articles made explicit references to treatment plans. This refined subset of articles will serve as a valuable resource for our analysis and examination of the relationship between values and treatment plans.

Through meticulous analysis, we categorized the treatment plans mentioned in the 341 selected articles, leading to the identification of 27 distinct combinations associated with specific values. This classification, visually represented in Figure 4, provides a comprehensive overview of the relationships between treatment plans and their corresponding values. To support our research and analysis, we compiled a comprehensive literature database consisting of article titles, abstracts, links, treatment plans mentioned in the articles, and their associated values. This database serves as a crucial foundation for constructing our value-based decision-making model. By conducting the manual reading of 100 papers from the selected subset, we extracted pertinent information that significantly influences decision making based on values, which has been thoughtfully incorporated into Table 3 as part of the literature database for values influencing decision making. The amalgamation of these steps and resources has enabled us to establish meaningful connections between treatment plans and the relevant values, contributing to an enhanced understanding of the impact of values on medical decision making.

### 2.4. A Unified Model for Decision Evidence

Through meticulous analysis of the extracted values from the decision literature database, as presented in Table 3, numerous pieces of evidence have emerged concerning the impact of values on decision making. Within the literature analysis process, it was revealed that the literature addresses the potential existence of single or multiple treatment options. Comparisons are drawn between these options, with a single value-based evidence potentially incorporating two treatment alternatives, one garnering support and the other facing opposition. The evidence that elucidates the influence of values on decision making encompasses the clinical context or population characteristics, pertinent values, the treatment options advocated or contested as well as the weight attributed to the evidence. Table 4 offers a comprehensive overview of decision justifications influenced by values, thus serving as an indispensable resource to guide value-based clinical decisions.

In order to facilitate the recognition and utilization of evidence by computers, surpassing the limitations imposed by tabular formats, an ontology named Unified Decision Evidences has been developed to organize the information. The upper part of Figure 5 illustrates the abstract ontology layer, wherein treatment options act as intermediate entities that establish connections between the clinical evidence ontology and the value-based evidence ontology. The relationships between these ontologies are exemplified in the lower part of Figure 5, employing endocrine therapy as a case study, amalgamating clinical evidence and the derived value-based evidence.

Upon the establishment of the ontology structure, an RDF framework has been devised to store and organize the argumentation structure showcased in the diagram. This RDF-based arrangement facilitates the seamless integration and linkage of information in a machine-readable format. The design of the argumentation structure aims to forge connections between arguments and diseases, thereby enabling a comprehensive representation of objective conditions and values. Furthermore, it offers a lucid depiction of treatment plans while furnishing supporting information to bolster the arguments.

Figure 6 portrays a prototypical argumentation structure encompassing five distinct parts. The first part establishes the correlation between the argumentation and the pertinent disease, thereby ensuring contextual relevance. The subsequent portion delineates the rules governing objective conditions, providing a framework for evaluating the patient’s objective medical status. The third component places emphasis on values, accentuating the pivotal subjective factors that influence the decision-making process. The fourth segment offers a detailed explication of the target treatment plan, specifying the viable courses of action to be considered. Finally, the fifth division supplies corroborative information, substantiating the arguments with pertinent evidence and references.

It is worth noting that in the “support_type” section of the fourth part, there are only two available values: “positive” and “negative.” When selecting medical research papers, we do not consider including cases where patient values have no impact on treatment plans in the knowledge repository. This neutral conclusion will not serve as evidence for patient-centered decision recommendations.

To activate these arguments, the patient’s information must align with the definitions of the disease, objective conditions, and values expounded in the argumentation. In the case of value-based arguments, specific criteria must be met with regard to the patient’s values. The activated arguments are subsequently compiled and subjected to statistical analysis to extract treatment support types and ascertain their impact. It is worth mentioning that the activation of value-based arguments requires the fulfillment of conditions based on the values possessed by the patient.

In order to enhance the clarity of unified decision evidence for use in patient healthcare decision making, we have described the UMDE decision inference model in Figure 7. The model takes the patient’s disease as input and invokes the corresponding knowledge base. The information is classified into clinical information and value-based information. By matching the clinical decision evidence with the evidence of values influencing the decision, we obtain an activated evidence library specific to the patient. Through the calculation of weights assigned to each treatment option mentioned in the evidence, we ultimately derive a ranked list of personalized treatment candidates for the patient along with the corresponding evidence set as the basis for the decision-making process.

### 2.5. Population-Based Value Pre-Configuration

This correlated general medical knowledge plays a vital role in the formulation of personalized treatment plans that are aligned with patients’ needs and values, leading to improved satisfaction, enhanced recovery, and overall health benefits (refer to Figure 8). The values of individuals are closely associated with their specific characteristics. For example, individuals in the public eye may prioritize appearance and charisma, while writers may highly value their creative abilities, and assembly line workers may prioritize rest. In our research, we specifically focus on exploring the relationship between occupation-related characteristics and values within the population. We selected five evenly distributed community service centers in the local city area and conducted a one-week survey activity in each community. We conducted a uniform survey of individuals aged 18 and above but below 65 to understand their occupations and explored their valued beliefs using interviews and questionnaires. During the interviews, we guided them to express the values related to their professions and asked them to rate the importance on a scale from 0 to 1. After completing the surveys in the five community service centers, we grouped data from individuals with the same occupations and calculated the average values. Ultimately, these values are used as the predefined weight values for the value system of different occupational groups. We employ statistical mapping techniques to identify the correlations between occupations and values, considering the impact of side effects on these values. Moreover, we provide preset values that represent the significance of such impact, contributing to a more comprehensive understanding of the values influencing decision-making processes.

The statistical preset values initially determined are further fine-tuned based on individualized values for each patient, thereby providing a more precise assessment of the influence of side effects on patients’ demographic characteristics. This personalized adjustment enables a better alignment of the preset values with the specific circumstances of each patient. For instance, personalized modifications can be made to accommodate patients in particular occupations, ensuring that the preset values are more tailored to their unique needs. Table 5 exemplifies preset values for three distinct occupations, serving as a valuable reference for healthcare professionals to consider patients’ demographic characteristics and values during the formulation of personalized treatment plans. By incorporating these considerations, doctors can provide more targeted and effective care that accounts for the diverse needs and values of their patients.

Based on these works, our approach of correlating population and values enables a comprehensive understanding of patients’ unique requirements, allowing us to effectively address their individualized needs. By incorporating this knowledge, we can develop treatment plans that are closely aligned with their values, resulting in more personalized and patient-centered care. This approach provides robust support for enhancing the quality of medical interventions and promoting patient satisfaction and well-being.

### 2.6. Recommendation Algorithm and Solution Output

Referring to Algorithm 1, we carefully consider both the clinical and value information provided by the patients. Initially, we matched the patients’ set of clinical symptoms with the set of arguments to identify the arguments that meet the specified conditions. Subsequently, the weight assigned to each matching argument is incorporated into the corresponding treatment plan mentioned within that argument. In the case of a positive treatment plan type, the recorded impact value is directly aggregated. Conversely, for a treatment plan with negative type, the recorded impact value is subtracted. By diligently following this process, we obtain a set of weights for each treatment plan, which are determined based on the objective symptom evidence. The specific algorithm implementation is shown in Algorithm 1.

Following that, we proceed to traverse the arguments and assess the patients’ set of values along with the objective symptom evidence that influences the weight of these values. To activate an argument, both the objective symptoms and values mentioned within the argument must be satisfied simultaneously. The information contained in the activated argument is then carefully analyzed. In the case of a treatment plan with a negative type, the weight impact is determined by multiplying the weight of the value involved in the corresponding argument within that treatment plan. Conversely, for a treatment plan with a positive type, the weight impact is directly added to the weight of the treatment plan. Finally, the weight ranking of the various treatment plans for the specific disease is established and presented as the outcome.
**Algorithm 1** An algorithm for calculating the weight of treatment plans that integrates values through patient information and arguments
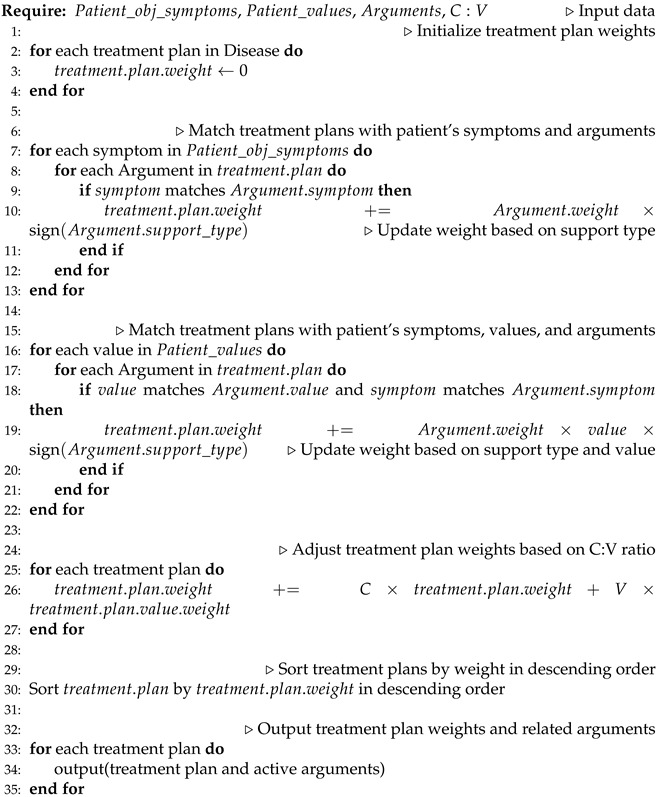


## 3. Result

### 3.1. Prototype System Implementation

To evaluate the efficacy of our methods and models, we developed and implemented a value-incorporated clinical decision system (VICDS). The system is designed to collect clinical information and values from patients with clinical information stored in key–value pairs to create a comprehensive collection of patient clinical data. By employing a series of guided questions, we obtain patients’ clinical information, occupational and personalized values. The patients’ value list comprises a predefined set of occupational values, which are each assigned a weight ranging from 0 to 1. Both clinical and value information can be added or adjusted by the patients, ensuring that the system captures information that better reflects their specific circumstances, thereby enabling more tailored recommendations. In our recommendation module, we have introduced sliders for patients and doctors, enabling them to adjust the weights and proportions of clinical and value perspectives. This empowers them to tailor the overall balance according to their individual needs. The effectiveness of the model is demonstrated in Figure 9 and Figure 10, where we showcase information from two patients with identical clinical data but different value profiles.

In the left part of Figure 9, we present the input information of a patient named Lisa, who is a star. Lisa highlights her busy schedule, emphasizing the limited time available for hospital treatments. Additionally, she expresses her desire for a future pregnancy and the importance of avoiding permanent infertility. Considering her occupation, Lisa places significant value on commercial performances and prioritizes her appearance. Moreover, Lisa demonstrates a higher threshold for pain compared to others and is willing to endure some discomfort in exchange for other values.

In the left part of Figure 10, we present the input information of a patient named Mata who works as an assembly worker. Mata emphasizes her preference for a favorable recovery outcome and is less concerned about the duration of the treatment period; her primary goal is to achieve a better recovery. She expresses significant fears regarding complications. Additionally, Mata hopes that the treatment plan will not incur excessive costs, as this would result in financial strain for her. Given her daily work involving prolonged movement and walking, it is crucial for Mata to avoid experiencing pain throughout the day, as it would have a substantial impact on both her personal and professional life.

Once the patients’ clinical and value information is collected, the matching of arguments is performed using Algorithm 1, resulting in sets of activated arguments and their corresponding weights. The process is illustrated in the right part of Figure 9 and Figure 10. The ratio between clinical weights and value weights can be adjusted through discussions between doctors and patients. Based on these adjustments, specific treatment recommendations are provided. The treatment plan is divided into three stages: pre-surgery, surgery, and post-surgery. Recommendations for each stage are derived from the arguments, with the default selection being the treatment plan with the highest weight. In the section explaining the treatment plan, the corresponding arguments and their weights are presented, providing doctors and patients with insights into the reasons behind the recommendations.

### 3.2. Model Validation and Evaluation

We invited 90 patients, 5 expert doctors from hospitals, and 10 medical students to participate in our evaluation experiment. We prepared a CDSS that considers only the objective symptoms of patients (called an OCDSS). The OCDSS relies solely on clinical guidelines to analyze patients’ objective symptoms and recommend treatment plans. Both the OCDSS and VICDS share the same clinical evidence database. The 90 patients were evenly divided into three groups, where the expert doctors made treatment decisions for the patients using traditional medical methods (called a DOCTOR), OCDSSs, and VICDSs, respectively.

In order to facilitate the evaluation of three treatment recommendation models, we conducted a preliminary survey of each volunteer before the experiment and recorded their clinical expectations for treatment outcomes. Since treatment is a long-term process, we used two survey questionnaires to collect patient satisfaction, and the patient’s primary physician also completed two satisfaction questionnaires at different stages of treatment. The specific survey questionnaires are shown in the text Box 2 below, including two types: one with scores ranging from 0 to 1, and the other collecting information in text form. We collaborated with 10 medical students to evaluate whether the patient’s current treatment situation has met or exceeded the expected results.
Box 2Display of questionnaire items.Patient survey:Satisfaction level with the chosen decision-making plan (values ranging from 0 to 1)What were the unacceptable situations during this treatment? (text)Which values were threatened? (text)Which values were protected? (text)
Doctor survey:
Satisfaction level with the chosen decision-making plan (values ranging from 0 to 1)Smoothness of the current patient’s treatment (values ranging from 0 to 1)

### 3.3. Experimental Results

We obtained the statistical data from two stages of survey questionnaires, as shown in Table 6. We classified the satisfaction levels by stages and calculated the mean values. The proportion of values for meeting or exceeding treatment expectations was obtained through medical students. From the data, we can see that in the DOCTOR experimental group, patient satisfaction decreased in the second stage compared to the first stage. We believe this is because in the early stages of treatment, some treatment side effects did not appear, but in the second stage, these side effects gradually began to challenge the patients’ values, leading to a decrease in patient satisfaction. We also observed that patients’ dissatisfaction could impact their attending physicians.

In the OCDSS experimental group, patient satisfaction also decreased from the first stage to the second stage, but overall satisfaction was higher compared to the DOCTOR experimental group. Some doctors expressed that having clinical knowledge helped them become more aware, which was considered a positive experience.

We found that the VICDS experimental group had higher levels of satisfaction among both doctors and patients, and there was a significant improvement in achieving the expected treatment outcomes. Through interviews, we learned that incorporating values in clinical decision making allowed patients to better uphold their own values, make better-balanced decisions when considering interpretive arguments, and prepare mentally or practically in advance when certain values might need to be compromised. Expert doctors stated that although incorporating values increased the complexity of decision making, patient treatment compliance improved significantly, which was a major reason for the substantial improvement in treatment outcomes. Systematized knowledge helped doctors gather more information and make reasonable decisions in collaboration with patients.

The findings revealed a high consistency between VICDS recommendations and patients’ reselection of treatment plans. The VICDS takes into account both patients’ objective symptoms and their values, which influence their treatment decisions. Patients can weigh the affected values mentioned in the recommendations, helping them prepare for potential sacrifices. Some differences in recommendations were observed, which were attributed to certain patients’ values not being systematically validated in published papers, leading to deviations in the results. Additionally, the study’s scope may not have covered all of the relevant literature, which could also contribute to discrepancies. Patients’ actual treatment choices without using the VICDS tended to focus more on objective symptoms, overlooking the impact on values, potentially resulting in less suitable treatment plans.

## 4. Discussion

Currently, medical technology is highly advanced, offering numerous treatment options for various diseases. However, due to the long update cycles of medical guidelines, there is often a lack of deliberate collection of information related to patients’ values in these guidelines. Patient-centered treatment approaches have not been fully incorporated into medical guidelines, and only a small fraction of articles from Narrative Reviews are likely to be included in the guidelines.Therefore, we emphasize the importance of patient involvement in decision making and explore personalized preferences and expectations of patients regarding treatment choices. Our research aims to incorporate patients’ values into the clinical decision-making evidence, leading to the development of the VICDS. Through the VICDS, we collect and integrate information from patients, considering their potential challenges and hesitations, and quantitatively store this data. By matching real patient information with evidence containing values and utilizing algorithms, we comprehensively rank treatment options, shifting the focus of healthcare toward patients. This approach provides valuable supplements to medical guidelines concerning decision making based on values and offers support and evidence for clinical decisions.

We conducted the tracking and recording of the treatment processes for three groups of volunteer patients and collected data on satisfaction and treatment outcomes. The results show that using the clinical decision support system provides doctors and patients with systematic clinical evidence supplements. Compared to not using the clinical decision support system, there is a slight increase in patient and doctor satisfaction and improved clinical treatment outcomes. The inclusion of patient values shifts the decision-making center toward patients, and doctors can incorporate patients’ values into their considerations through system prompts. Patients find that using the VICDS for decision making allows them to weigh the evidence presented and make well-informed psychological preparations and lifestyle adjustments.

This study makes three main contributions with the aim of providing comprehensive and personalized support for decision making in specific disease domains.

Firstly, medical objective factors such as diseases and treatment plans are acquired from medical clinical guidelines. An online literature database is screened and analyzed using disease names, treatment plans, and values as keywords. Through this process, a Value-Based Evidence Database (VBED) is constructed, encompassing the relevant literature in the specific disease field where values influence decision making. The importance of values in the decision-making process is considered, ensuring that the database covers not only objective medical guidelines but also the impact of values.

Secondly, the evidence database of how values influence decision making is analyzed and formalized. The goal is to unify value-based evidence with objective evidence, establishing the Unified Model for Decision Making with Values (UMDE). Through this unified model, healthcare professionals and patients can better understand and balance different factors when making treatment choices.

Thirdly, a prototype system called the VICDS is designed and implemented. The VICDS matches patient information with the unified evidence and provides comprehensive and personalized treatment recommendations. Leveraging the information in the database and the structure of the model, the VICDS considers individual patient needs and values, offering customized advice for each patient. This approach aims to enhance the treatment experience for patients and facilitate better treatment outcomes.

In summary, this study contributes by constructing a Value-Based Evidence Database, unifying objective evidence with value-based evidence, and designing a prototype system that provides comprehensive and personalized treatment recommendations for patients. These contributions aim to provide healthcare professionals and patients with more comprehensive and accurate information for decision making in specific disease domains, thereby promoting better treatment decisions and outcomes. We acknowledge that the VICDS currently has certain limitations and shortcomings. Firstly, the influence of values on the decision-making evidence database is somewhat limited due to the current scarcity of evidence, with a predominant focus on breast cancer. This represents a notable constraint of the VICDS at present. Secondly, the diversity of population-based pre-configurations is restricted by the limited diversity in sampling and statistics across different regions, potentially introducing some region-specific biases. Moreover, the sample size may not be extensive enough, leading to a potential lack of precision in mapping the values of the population. To address this, continuous data acquisition is essential to enhance the accuracy of the pre-configurations. Lastly, in the final recommendation results page, the VICDS primarily emphasizes the overall ranking of treatment plans without providing a specific comparison of the advantages and disadvantages between individual plans.

## 5. Conclusions

This study uses disease names, treatment plans, and values as keywords for a literature analysis to construct a value evidence database in a certain disease field where values influence decision making. By analyzing the value evidence database and formalizing it, the goal is to unify it with objective evidence. The unified evidence is used to provide patients with comprehensive decision making.

The contributions of this article are as follows:The correlation of the factors that influence values with treatment plans was made, and evidence of patient values was established based on literature analysis.A unified formalism of value-based evidence and clinical evidence was developed.A running CDSS prototype for breast cancer was built on top of the unified formalism, and its feasibility and effectiveness were evaluated.

In the future, we will utilize more real-world cases to optimize our preset values. Simultaneously, we will refine our question–answer guidance through feedback from patients and doctors, fully exploring patient information to enhance user experience. Additionally, we aim to establish relevant knowledge graphs to explore the application of Markov Ontology Theory in calculating weights. We will explore the incorporation of value-based decision evidence into local treatment plans, but this may introduce fairness concerns in overall treatment plan rankings. Therefore, one of our future research goals is to explore alternative methods to mitigate the impact on fairness and seek further improvements.

Moreover, we also plan to explore the following directions in future research: firstly, identifying the relative importance of values in different disease domains and quantifying them concretely; secondly, investigating which disease domains can be validated earliest with the proposed methods; thirdly, providing a value-based evidence repository influencing decision making for more disease domains, applying the algorithm to a broader range of diseases, and validating its applicability and effectiveness across different diseases; and fourthly, exploring whether the methods and systems mentioned in this paper can be integrated with others to ultimately achieve more meaningful advancements in the medical field.

We will strive to diversify our projects further to refine and expand the proposed system prototype, providing comprehensive and effective support for future medical decision making and treatment choices.

## Figures and Tables

**Figure 1 healthcare-11-02362-f001:**
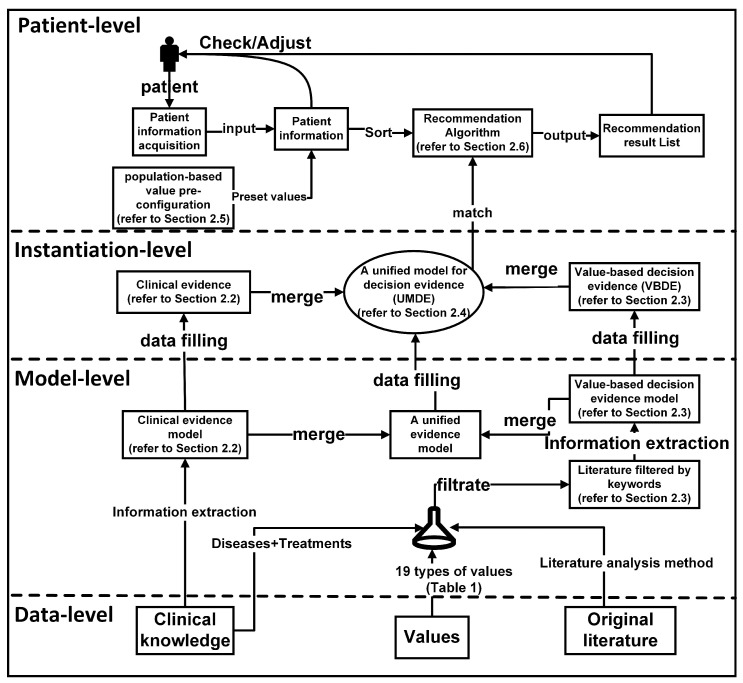
Overall flowchart and main components of this article.

**Figure 2 healthcare-11-02362-f002:**
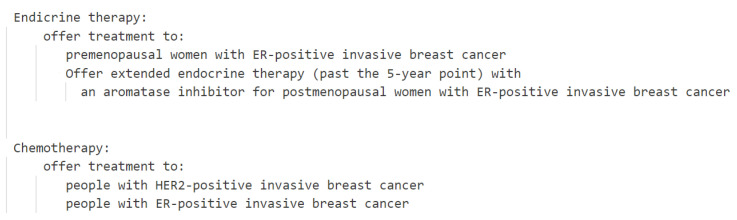
Examples of conditions and treatment plans in NICE.

**Figure 3 healthcare-11-02362-f003:**
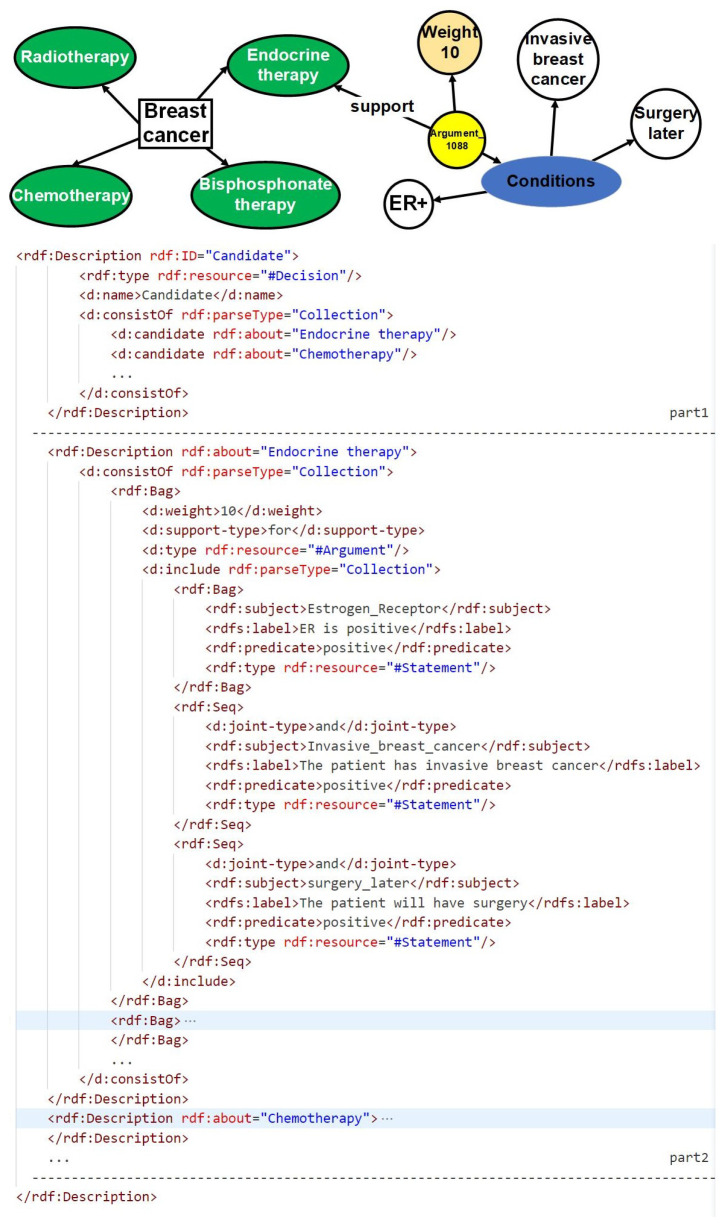
Clinical evidence model using breast cancer as an example.

**Figure 4 healthcare-11-02362-f004:**
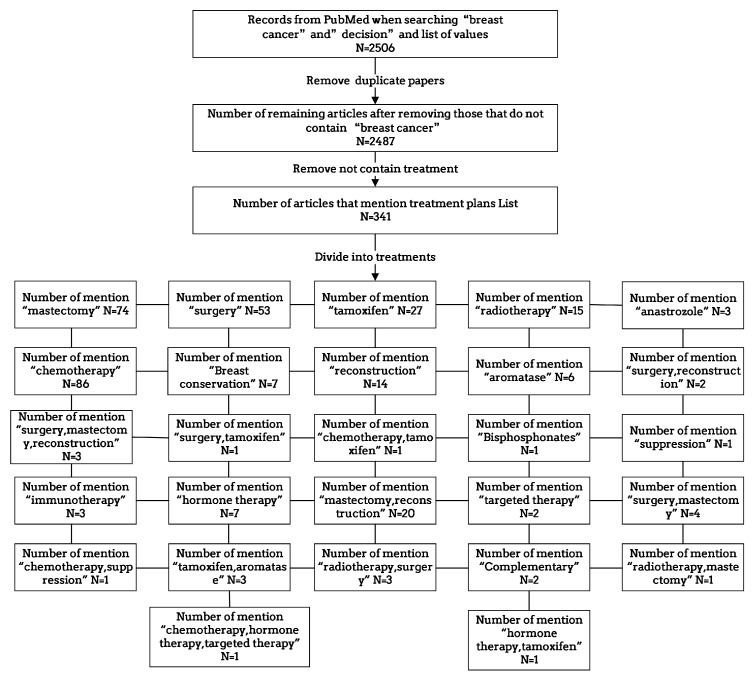
The process and quantity of literature analysis based on keywords.

**Figure 5 healthcare-11-02362-f005:**
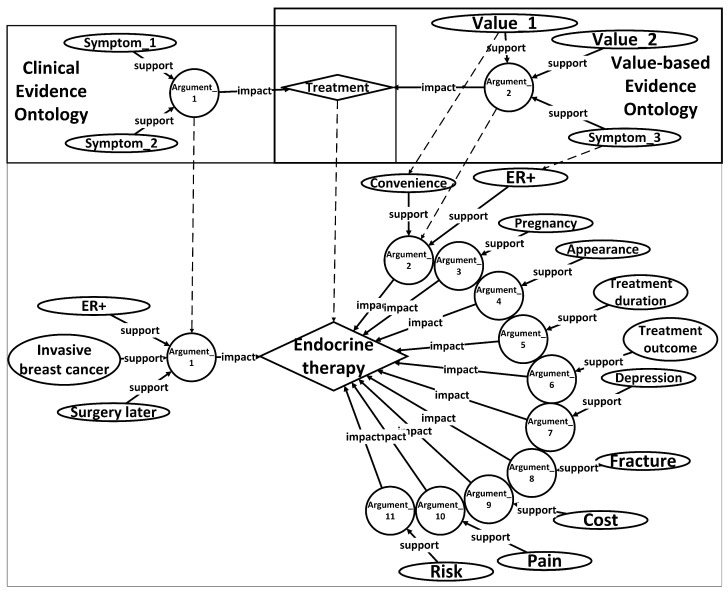
Ontology of an unified decision evidence.

**Figure 6 healthcare-11-02362-f006:**
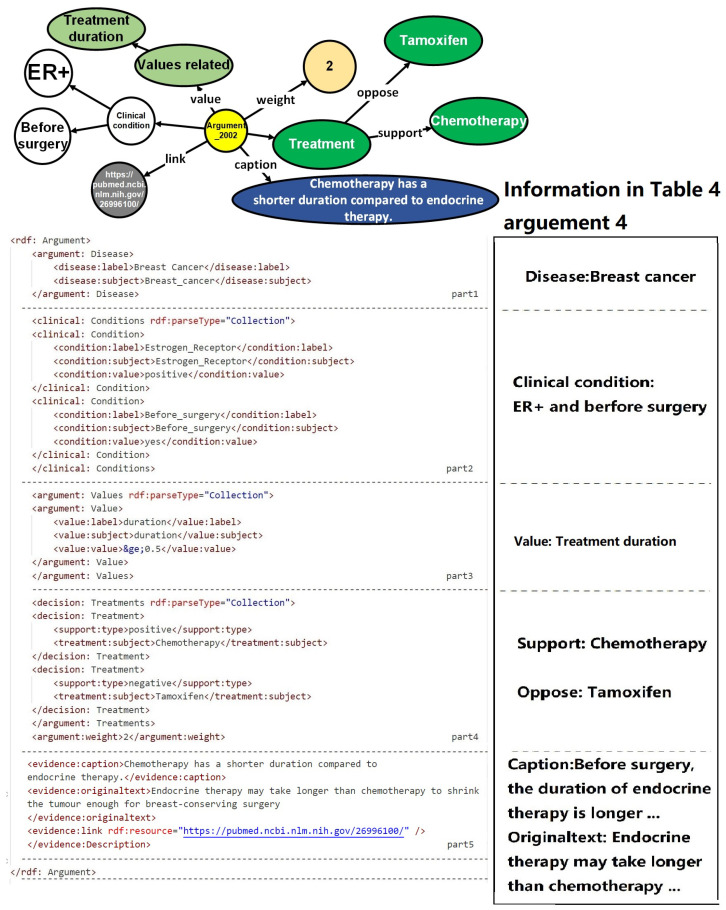
An example of a normalized argument stored in an RDF file for subsequent matching of patient information and recommendation of treatment plans.

**Figure 7 healthcare-11-02362-f007:**
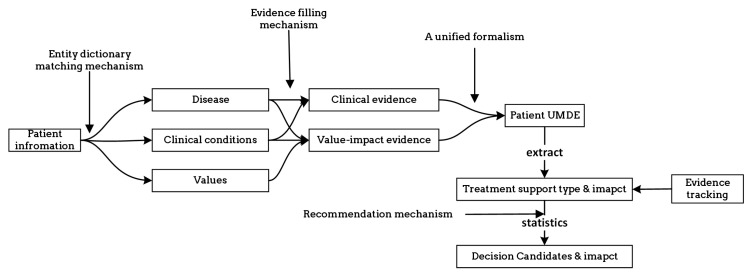
Description of the process by which patient information is transformed into decision candidates.

**Figure 8 healthcare-11-02362-f008:**
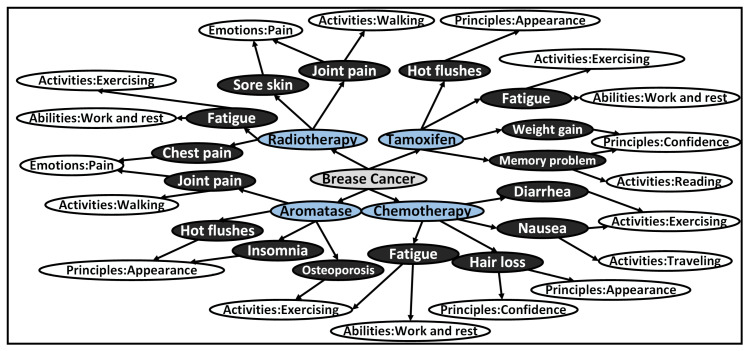
Structure that takes breast cancer as an example to show the mapping of general medical knowledge and values.

**Figure 9 healthcare-11-02362-f009:**
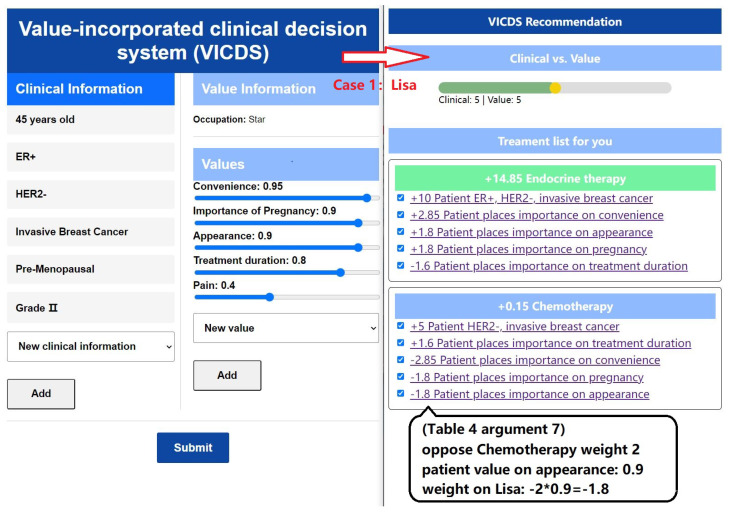
Case 1: Lisa’s information input and treatment plan recommendation display.

**Figure 10 healthcare-11-02362-f010:**
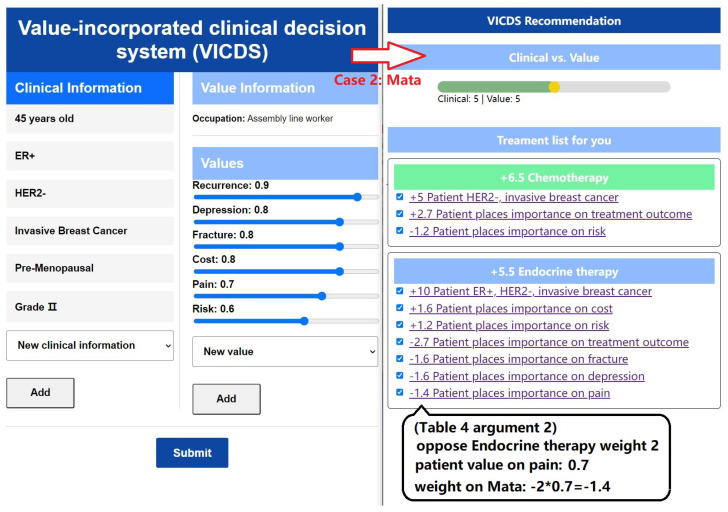
Case 2: Mata’s information input and treatment plan recommendation display.

**Table 1 healthcare-11-02362-t001:** Six categories of values and 29 fine-grained values.

Six Categories Values	Fine-Grained Values	Six Categories Values	Fine-Grained Values
Activities	fracture	Principles	independence
	recurrence		treatment duration
	traveling		confidence
	reading		appearance
	walking		weight
Abilities	survival	Emotions	risk
	work		pregnancy
	rest		exhaustion
	talking		pain
	vision		unbearable
	convenience		depression
	exercise		
Possessions	convenience	Relationships	family
	transportation		friend
	expensive		colleague
	cost effective		community

**Table 2 healthcare-11-02362-t002:** Keywords of relevant treatment plans found in NICE, taking breast cancer as an example.

No.	Category	Keywords
1	Endocrine therapy	Hormone therapy
		Tamoxifen
		Aromatase inhibitors
		Ovarian ablation/suppression
2	Radiotherapy	-
3	Chemotherapy	-
4	Surgery	Mastectomy
		Breast reconstruction
		Breast conservation
5	Targeted Therapy	-
6	Immunotherapy	-
7	Bisphosphonates	-
8	Complementary Therapy	-

**Table 3 healthcare-11-02362-t003:** Impact of values on decision making, which is extracted or analyzed from the literature.

Index	Author/Year	Title	Finding	Link Start with https://pubmed.ncbi.nlm.nih.gov/
1	Durrani/2020	Controversies Regarding Ovarian Suppression and Infertility in Early Stage Breast Cancer [20]	The main concern after adjuvant chemotherapy is the risk of losing fertility, as chemotherapy can induce early menopause in most premenopausal breast cancer patients. Tamoxifen only slightly increases the risk of early menopause.	32104064/
2	Castel/2013	Time course of arthralgia among women initiating aromatase inhibitor therapy and a postmenopausal comparison group in a prospective cohort [21]	Women undergoing endocrine therapy have more severe joint pain, and they have more severe menopausal symptoms or existing joint-related diseases relative to before treatment. Joint pain is more severe than expected after menopause and often leads to reduced compliance.	23575918/
3	Rachner/2018	Bone health during endocrine therapy for cancer [22]	Common osteoporosis guidelines are likely to have underestimated the fracture risk of patients receiving endocrine therapy—especially in patients on aromatase inhibitor therapy.	29572126/
4	Jankowitz/2013	Optimal systemic therapy for premenopausal women with hormone receptor-positive breast cancer [23]	Chemotherapy has a shorter duration compared to endocrine therapy.	26996100/
5	Murray/2006	Neoadjuvant endocrine therapy models [24]	Chemotherapy is more effective than endocrine therapy at shrinking the tumor.	16491621/
6	Collier/1997	New aromatase inhibitors for breast cancer [25]	Endocrine therapy can provide self-administered oral medication, while chemotherapy requires injections at the hospital.	9282426/
7	Kanti/2015	Evaluation of trichodynia (hair pain) during chemotherapy or tamoxifen treatment in breast cancer patients [26]	Chemotherapy has more severe hair loss and scalp pain compared to Tamoxifen, and the duration is also longer.	26403680/
8	Reinert/2018	Current Status of Neoadjuvant Endocrine Therapy in Early Stage Breast Cancer [27]	Endocrine therapy is a practical, cost-effective treatment.	29663173/
9	Lima/2017	Temporal influence of endocrine therapy with tamoxifen and chemotherapy on nutritional risk and obesity in breast cancer patients [28]	Women on endocrine therapy with TMX are mostly overweight and obese, most evidently in women who received CT and those who were at the beginning of treatment.	28851304/
10	Desai/2021	Breast Cancer in Women Over 65 years—a Review of Screening and Treatment Options [29]	Primary endocrine therapy is a low-risk option for those with limited life expectancy.	34600726/
11	Brown/2020	Post-traumatic stress disorder and breast cancer: Risk factors and the role of inflammation and endocrine function [30]	Tamoxifen also has been shown to be involved in adverse mood reactions such as depression.	32374431/
12	Huang/2023	Cost-effectiveness analysis of ovarian function preservation with GnRH agonist during chemotherapy in premenopausal women with early breast cancer [31]	GnRHa plus Chemo was a cost-effective strategy for premenopausal women with BC in the USA.	37075316/
13	Eills/2006	Initial versus sequential adjuvant aromatase inhibitor therapy: a review of the current data [32]	For those with positive nodes, the initiation of treatment with aromatase inhibitors may be beneficial to avoid tamoxifen-associated early relapses after diagnosis.	17257462/
14	Eills/2006	Initial versus sequential adjuvant aromatase inhibitor therapy: a review of the current data [32]	From an economic perspective, aromatase inhibitors are considered cost-effective compared to tamoxifen.	17257462/
15	Lee/2019	Association between C-reactive protein and radiotherapy-related pain in a tri-racial/ethnic population of breast cancer patients: a prospective cohort study [33]	In the postoperative radiotherapy process of obese patients, pain occurs, which has a negative impact on the quality of life.	31138314/
16	Hodis/2008	Postmenopausal hormone therapy and cardiovascular disease in perspective [34]	Hormone therapy after menopause can reduce the mortality rate and the risk of coronary heart disease.	18677151/

**Table 4 healthcare-11-02362-t004:** Impact of values on decision making where the information is extracted and analyzed from literature.

Index	Clinical Condition	Value	Support	Oppose	Weight	Source in Table 3
1	premenopausal	pregnancy later	Endocrine therapy	Chemotherapy	2	1
2	-	pain	-	Endocrine therapy	2	2
3	-	fracture	-	Endocrine therapy	2	3
4	ER+, pre-surgery	treatment duration	Chemotherapy	Endocrine therapy	2	4
5	premenopausal	treatment outcome	Chemotherapy	Endocrine therapy	3	5
6	ER+, HER2-	convenience	Endocrine therapy	Chemotherapy	3	6
7	-	appearance	Endocrine therapy	Chemotherapy	2	7
8	ER+, Grade 2	cost	Endocrine therapy	-	2	8
9	-	weight	-	Endocrine therapy	2	9
10	ER+, age > 65	risk	-	Endocrine therapy	2	10
11	ER+	depression	-	Endocrine therapy	2	11
12	age 18–49, premenopausal	family cost effective	Chemotherapy+ GnRHa	Chemotherapy	3	12
13	node-positive	recurrence	Anastrozole	Tamoxifen	2	13
14	-	cost effective	Aromatase	Tamoxifen	2	14
15	age < 50, after surgery, overweight	pain	-	Radiotherapy	2	15
16	postmenopausal	mobility, survival	-	Hormone therapy	2	16
17	ER+	risk	Endocrine therapy	Chemotherapy	2	5

**Table 5 healthcare-11-02362-t005:** Population-based value pre-configuration.

Population Features	Related	Side Effects	Weight
Actor	Appearance, Temperament	Weight gain	0.88
		Alopecia	0.85
		Skin darkens	0.74
		Diarrhea	0.30
Writer	Creativity, Spirit	Memory loss	0.94
		Tremor	0.87
		Insomnia	0.81
		Fatigue	0.67
Assembly line worker	Work and rest, Repetitive work	Joint pain	0.86
		Insomnia	0.73
		Numbness in limbs	0.70
		Back pain	0.65

**Table 6 healthcare-11-02362-t006:** Comparative display of the results of two information collections.

	First Information Collection			Second Information Collection		
Statistical Items	DOCTOR	OCDSS	VICDS	DOCTOR	OCDSS	VICDS
Patient satisfaction	0.81	0.87	0.95	0.76	0.88	0.95
Doctor satisfaction	0.93	0.87	0.91	0.93	0.89	0.93
Outcomes met or exceeded expectations	0.43	0.50	0.56	0.70	0.76	0.86

## Data Availability

The data presented in this study are available on request from the corresponding authors.

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
