# Peer review of "Toward a Value-Based Therapy Recommendation Model"

_healthcare, 2023, doi:10.3390/healthcare11162362_

Round 1
Reviewer 1 Report
In this highly intriguing article, the authors present a pilot study on the modeling of treatment recommendations based on patients' values. The topic addressed is exceedingly interesting and provides an innovative perspective on an unresolved issue in the field. However, there are identified areas for improvement regarding the format, structure, and content of the article, for it to be considered for publication.
Mayor issue:
1) The present manuscript deviates from the established structure of a scientific article or review; instead, it amalgamates elements from both. On careful examination, the authors advise to align the manuscript's structure more closely with that of a scientific article. Specifically, I suggest condensing points 1 and 2 into a unified introduction that culminates in a clear statement of the study objective in the final paragraph. The introduction can be further streamlined by presenting two or three key ideas while also providing a concise enumeration of pertinent previous works.
2) The objectives of the study must be reviewed and simplified.
Objective 1. What exactly are patients' values that affect clinical decision-making and how can we build value-based evidence for it? This question is trying to answer based on the result of the narrative review. Point 3.3 Materials and Methods.
Narrative reviews, while informative and insightful, have certain limitations that need to be acknowledged. One primary concern is the potential for bias in the selection and interpretation of the included studies, as they lack a systematic and standardized approach to search and appraisal. Additionally, the absence of a predefined methodology and rigorous data synthesis may compromise the overall reliability and reproducibility of the findings. Systematic reviews employ a rigorous and transparent methodology to synthesize existing evidence on a specific research question or topic. They follow a well-defined protocol that outlines the objectives, inclusion criteria for studies, search strategy, data extraction methods, and statistical analysis, if applicable.
I suggest clarifying the methodology of the review, summarize the findings, or include in supplementary materials, and remove it from the objectives. The authors chose one method of building a valued-based evidence based on a narrative review but is not a novelty.
Objective 2 and 3. Can we put forward a unified formalism for value-based evidence and clinical evidence already adopted, so that decision makers can be supported in a systematic manner? Evaluating the model of its feasibility, effectiveness, etc.
This is the real objective of the study. The construction and validation of an algorithm to support the clinical decision process based on patient value. I suggest not writing it in a question, but more appropriate writing it in a sentence concise and clear.
Results
3) GPT chat is not a valid validation system. In your result you are compared your algorithm with other algorithms but that is not a validation technique. If the algorithm modifies the treatment option base on patient values, ask in an external cohort of patient about their treatment options and compare it with the algorithm prediction.
Discussion
4) The discussion appears to be significantly concise and lacking in comprehensive analysis. It is recommended to incorporate a comparative analysis that encompasses alternative perspectives on treatment selection based on patient values, predictive efficacy, or model goodness of fit. Notably, the most salient concern stemming from the brevity of the discussion, and one of particular significance, lies in its detachment from the patient's viewpoint. Given that the algorithm is supposedly based on patient values, a dedicated paragraph elucidating the patient-centred perspective would have been essential to address this aspect thoroughly.
Minor issue.
5) The NICE guideline used has not been referenced, and articles selected in the narrative review could potentially be included in the guideline. Please review and indicate if this occurs.
6) The article contains an excessive burden of tables, figures, and diagrams. I suggest including some as supplementary material.
7) In line 425, the expression "work as star" appears, which is not correct. Please review and modify it.
8) References 9-10 and 15-16 appear to be instances of self-citations, as they represent the same communication initially presented at a conference and subsequently published in a journal. It is recommended to carefully assess these references and, should they pertain to identical works, exercise caution to avoid an undue abundance of self-citation. The overuse of self-citations has the potential to diminish the credibility of both the researcher and the journal. Therefore, prudent consideration of self-citing practices is essential to uphold the integrity and objectivity of scholarly publication.
9) In the middle of the article, on lines 381-386, the phrase "In conclusion" appears. The conclusion represents the final section of an article, substantiated by the results, and discussed in the preceding sections. Kindly review and modify the wording of the paragraph accordingly.
Reviewer 2 Report
The work is very interesting. The modeling is quite analysed
Recommentation.
it will be interesting the calculation of the weights for model Fig 5 consideirng markov ontologies theory
In table 5 is not very clear how the weights are calculated
Reviewer 3 Report
In this paper, authors proposed a value-based evidence database, integrated objective evidence with value-based evidence, and designed a prototype system (VICDS) that provides comprehensive and personalized treatment recommendations to patients, and verified their effectiveness. This study is a very comprehensive and in-depth study and is a very valuable study for patient management and treatment.
In order to increase the completeness of this paper, please revise the following.
- The abstract is relatively simple compared to the volume of the entire paper. Please describe the contribution of this study in more detail in the abstract.
-Describe the clear definition of patient value in Chapter 1.
-What is the clear meaning of 'fuse' in Figure 1 and it would be nice to replace it with other clear word.
-Can't Argument.support_type have a value of 0 for algorithm 1?
-Also, if possible in Algorithm 1, try to remove the conditional statement and describe it simply.
-It would be better to describe it in comment form in Algorithm 1 instead of Table 6.
-Describe the limitations or shortcomings of the prototype (VICDS) proposed in this study in Chapter 5.
-The future research issues described at the end of Chapter 6 take it too natural because the system proposed in this study is a prototype. Therefore, describe more diverse future research projects.
Round 2
Reviewer 1 Report
Dear authors, the article has changed substantially and is now clearer, but I still recommend a small change in the structure. You should include point 2 within point 1 as an introduction and remove heading 2. At the end, include the objectives (what is currently in lines 62-69). Lines 70 to 81 should be deleted as they do not contribute anything. Finally, rename point 4 (new point 3) as Results.
